# Insights from Self-Organizing Maps for Predicting Accessibility Demand for Healthcare Infrastructure

**Jerome R. Mayaud** [1,*] **, Sam Anderson** [2] **, Martino Tran** [1] **and Valentina Radić** [2]

1 School of Community and Regional Planning, Faculty of Applied Science, University of British Columbia, 6333 Memorial Road, Vancouver, BC V6T 1Z2, Canada; martino.tran@ubc.ca
2 Department of Earth, Ocean and Atmospheric Sciences, Faculty of Science, 2020–2207 Main Mall, Vancouver, BC V6T 1Z4, Canada; sanderson@eoas.ubc.ca (S.A.); vradic@eoas.ubc.ca (V.R.)
* Correspondence: jerome.mayaud@gmail.com

**Abstract:** As urban populations grow worldwide, it becomes increasingly important to critically analyse accessibility—the ease with which residents can reach key places or opportunities. The combination of 'big data' and advances in computational techniques such as machine learning (ML) could be a boon for urban accessibility studies, yet their application in this field remains limited. In this study, we provided detailed predictions of healthcare accessibility across a rapidly growing city and related them to socio-economic factors using a combination of classical and modern data analysis methods. Using the City of Surrey (Canada) as a case study, we clustered high-resolution income data for 2016 and 2022 using principal component analysis (PCA) and a powerful ML clustering tool, the self-organising map (SOM). We then combined this with door-to-door travel times to hospitals and clinics, calculated using a simple open-source tool. Focusing our analysis on senior populations (65+ years), we found that higher income clusters are projected to become more prevalent across Surrey over our study period. Low income clusters have on average better accessibility to healthcare facilities than high income clusters in both 2016 and 2022. Population growth will be the biggest accessibility challenge in neighbourhoods with good existing access to healthcare, whereas income change (both positive and negative) will be most challenging in poorly connected neighbourhoods. A dual accessibility problem may arise in Surrey: first, large senior populations will reside in areas with access to numerous and close-by, clinics, putting pressure on existing facilities for specialised services. Second, lower-income seniors will increasingly reside in areas poorly connected to healthcare services, which may impact accessibility equity. We demonstrate that combining PCA and SOM clustering techniques results in novel insights for predicting accessibility at the neighbourhood level. This allows for robust planning policy recommendations to be drawn from large multivariate datasets.

**Keywords:** smart cities; machine learning; big data; clustering; principal component analysis (PCA)

## 1. Introduction

Accessibility (the ease with which people can reach places or opportunities) is an important indicator of the liveability and sustainability of cities. In the global drive to make cities 'smarter,' there has been an observable trend for greater urban interconnection, especially through investments in physical infrastructure and information communication technologies (ICT) [1,2]. Innovative networks of connected devices and sensors, coupled with smart applications and data analytics, have also revolutionised the ways in which a variety of actors—from city governments to citizens—can simultaneously optimise existing systems [3], improve the quality of life urban residents [4,5] and address sustainable development needs [6–8]. There are widespread calls for smart cities to ensure their

services remain inclusive and equitable as they grow [9,10], because evidence shows that universal access to basic services positively impacts human development and societal progress [11–14].

However, disparities persist in residents' access to essential services across the development spectrum. Accessibility is therefore often investigated through the lens of equity, which can be defined as the provision of opportunities for individuals to enjoy healthy and fulfilling lives [15]. Access to healthcare is a fundamental right in many countries [16], as it plays an essential role in satisfying many of people's basic needs. Disparities in healthcare access arise because of a variety of physical and socioeconomic factors, including social status, household income, transport infrastructure and the spatial distribution of cities [17,18] but it is difficult to conclusively isolate causation within complex urban fabrics [19]. Poor healthcare accessibility is known to result in lower healthcare utilization and inferior health outcomes [18,20,21], so determining where to build new healthcare facilities is an ongoing challenge for urban planners. Moreover, declining fertility rates and increased longevity are resulting in older, less mobile populations with different accessibility needs compared to younger, more mobile ones [15,22]. A major issue for growing cities is to ensure that their transportation systems remain inclusive and equitable across different population groups [23].

Previous studies have attempted to link urban accessibility patterns to demographic and socioeconomic factors at various scales. These range from global accessibility towards urban centres [24], to accessibility to sporting infrastructure at the megacity scale [25] and access to essential services at the neighbourhood scale [26]. A common approach in these studies is to derive statistical relationships between the basic characteristics of catchment populations. However, whilst valuable insights can be drawn from such analysis, basic descriptors such as median or grouped income do not always provide sufficient nuance for in-depth analysis of accessibility changes across populations.

The increasing prevalence of 'big data' in the urban sphere makes it possible to observe, analyse and predict human behaviour at increasingly fine scales [27–29]. Advances in computational data mining based on machine learning (ML) techniques are also enabling researchers to make better sense of these rich datasets [30]. When such methods are 'unsupervised,' they imply that no a-priori knowledge of patterns is necessary [31]. Algorithms that can self-learn in this way are useful for dealing with data that have not been classified, including in cases where an underlying probability density function has to be estimated from observed data [30].

A powerful unsupervised ML algorithm is the self-organizing map (SOM), which uses artificial neural networks to characterise variability and summarise key features in a dataset [32]. SOM algorithms have been applied to a wide range of disciplines (see reviews in Reference [33]), from analysis of volcano seismic spectra [34] and atmospheric aerosol tracking [35] to land value prediction [36]. Other ML techniques (e.g., neural networks, random forest classifiers) have also been applied to urban transportation problems, including for predicting traffic flows [27] and travel mode choice [37] and for clustering transit card usage [38]. However, barring some exceptions (e.g., [39–42]), SOM analysis has had a relatively limited impact in the social sciences, despite the possibilities it offers to intuitively visualise complex socio-demographic patterns, which can notably be helpful in the context of policy-making [42,43].

In this study, we aim to relate socio-economic factors to accessibility across a city experiencing rapid population growth. We explore the potential effects that changing spatial distributions of income over time (2016–2022) may have for accessibility to healthcare services in the City of Surrey, Canada. We use Surrey as a representative case study of many modern, affluent cities around the world that are facing challenges associated with population change and urban development. We focus our analyses on seniors (65+ years of age), as they have more complex healthcare needs than the average population and therefore require attention when formulating health and transport policies [44]. Building on the study of Mayaud et al. [26], who analysed healthcare and school catchments in Surrey, we introduce a methodology that combines two clustering approaches to explore income distributions in space and time: (i) principal component analysis (PCA) and hierarchical clustering and (ii) SOMs. We use both approaches to reduce the dimensionality of our income datasets and draw out patterns in the

underlying data, to then relate them to accessibility metrics. While both PCA and SOM are established methodologies in themselves, our combination of classical and modern data analysis methods provides novel, detailed insights into a city's evolution in terms of population growth, income distribution and accessibility to essential services.

The remainder of this paper is presented in three parts. The next section introduces the socio-economic context of our case study, outlines the data sources used and explains the use of PCA and SOM in this study. Section 3 presents our analysis and a discussion of the results. Section 4 summarizes the main conclusions of this paper.

## 2. Methods

### 2.1. Study Area

The City of Surrey is one of the 23 local authorities in the regional district of Metro Vancouver and one of the largest cities in British Columbia with 520,000 inhabitants. The city's population has increased by ~40% since 2000 and its projected growth to ~800,000 by 2041 means it is the fastest growing city in the region [45]. Whilst the proportion of seniors increased from 10% to 14% over the last twenty years, the proportion of children and youth (0–19 years) has fallen steadily from 30% to 24% in the same period [46].

### 2.2. Data Sources

In order to emphasise the transferability of our analyses in this study, we relied solely on open data sources and open-source code. We assessed population characteristics (count, income and age) of Surrey's population, using Canadian census data collected at the smallest standard census geographic level, the Dissemination Area (DA) [46]. Census data for 2016 (the most recent complete census dataset available) and projected data for 2022 (the latest high-resolution projected data available to us), were acquired via the DemoStats database [47]. The 2022 projections—which should be viewed as only one of a possible range of future population scenarios—were compiled by DemoStats based on a combination of econometric, demographic and geographic models [48].

The DA-level census data were reorganized into a grid covering the entirety of the City of Surrey's boundaries, which was composed of 1,480 equally sized hexagons of a diagonal diameter of 500 m (0.16 km$^2$). The hexagonal shape of the cell was chosen to reduce sampling bias from edge effects and the size of the grid cell was designed to provide enough granularity in our spatial data while roughly matching the average size of a DA in Surrey. This helped to minimise downscaling errors when assigning census data to hexagonal cells. We followed the method of [26] to assign the census data to each hexagonal cell (see Section 1.1 of the Supplementary Materials for a brief summary).

Location data for healthcare facilities (walk-in clinics and hospitals) were obtained from the Open Data catalogues of British Columbia and Surrey. In total, we acquired location data for two hospitals and 33 walk-in clinics. Section 1.2 of the Supplementary Materials discusses how hospitals and clinics were defined and chosen in this study.

OpenStreetMap (OSM) was used to determine the spatial layout of pedestrian infrastructure and road networks. This formed the basis for the door-to-door routing algorithm that determined travel times (see Section 2.6). OSM is a prominent example of volunteered geographic information, where users contribute voluntarily to its open-source development. As such, some questions have arisen about its accuracy in some parts of the world. In a global analysis of the 'completeness' of OpenStreetMap, Barrington-Leigh & Millard-Ball [49] found that Canada has a fully mapped street network, so we consider OSM to be reliably complete for the analyses we present here.

Geolocated timetable data of public transportation routes and stops were acquired for September 2017 from Translink, the regional transit operator. These data are organized in the format of General Transit Feed Specification (GTFS), which represents transit timetabling but not dynamic traffic congestion.

### 2.3. Data Preparation

We seek to analyse the relationship between income distributions and accessibility metrics across the City of Surrey. Similar to [50], we do this by first reducing the dimensionality of our income distribution datasets through PCA. This approach identifies features that capture the most variability in our income distribution datasets. We then apply our SOM algorithm to the income data.

In order to perform PCA and SOM techniques on categorical census data, we first estimate the undefined upper bound of the income distribution using Pareto's Law of Income Distribution (see Section 1.3 in the Supplementary Information). For all DAs in 2016, the midpoint of the last, open-ended income category was calculated as $266,950. We then convert the categorical data to a continuous data distribution by assigning a basic midpoint estimator for each income bin and duplicating each estimator $Z$ number of times into a vector, where $Z$ is the frequency of occurrence of the bin in question [51]. We use MATLAB's built-in '*ksdensity*' kernel smoothing method to perform a kernel density estimation (KDE), which is a well-established technique for estimating probability density functions [52–55]. The KDE for each income vector is demeaned and normalized by its standard deviation. This provides us with a smoothed, normalised continuous probability distribution for income in each grid cell, which is used as the input for our PCA and SOM algorithms.

### 2.4. Principal Component Analysis and Hierarchical Clustering

Principal component analysis (PCA) is a classical data analysis method for reducing dimensionality of a high-dimensional dataset. Following [34], we align our nomenclature for PCA concepts with [56]. In PCA, a high-dimensional dataset is decomposed into a set of linearly independent 'modes' (also known as 'eigenvectors'). These modes are basis functions that span the space of the original dataset and are chosen to maximize the amount of variance explained by each mode [57]. The original dataset can be reconstructed by summing each mode weighted by a 'principal component' (PC); in other words, PCs are the projections of the data onto the modes:

$$d_i = \sum_{j=1}^{N} PC_{ij}.e_j \tag{1}$$

where $d_i$ is the dataset, $e_j$ is the mode, $PC_{ij}$ is the principal component and $N$ is the total number of modes. Alternatively, the original dataset can be reconstructed using only the first few (most important) modes, ensuring that a majority of the variance is accounted for whilst filtering out noise:

$$d_i = \sum_{j=1}^{M} PC_{ij}.e_j \tag{2}$$

where $M$ is the number of chosen modes, such that $M \ll N$.

In order to compare how characteristic income distributions change spatially between 2016 to 2022, we first perform PCA on the 2016 income distribution data. This decomposes the dataset into modes and PCs that help to capture where the most variability occurs within the dataset. Next, we cluster each 2016 KDE in the space of the first three PCs using hierarchical clustering. We use only the first three modes because they contain >97% of the variance in the 2016 dataset (i.e., we group income distributions according to how similarly they are reconstructed by the first three modes; see Section 3.1). Dendrograms are used to identify an optimal number of clusters [58]. The characteristic cluster patterns are constructed by using the mean PCs of the members of each cluster. Finally, we calculate the root mean squared error (RMSE) between each 2022 income KDE and each 2016 cluster pattern and assign each 2022 KDE to the cluster that results in the minimum RMSE. In this way, we group the 2022 income distributions based on their similarity to the observed 2016 cluster patterns, with RMSE used as a similarity metric.

In this study, PCA is used to ascertain that SOM cluster outputs are realistic. We acknowledge the qualitative nature of using a prescribed clustering algorithm to compare with the unsupervised SOM. Some studies have sought to more robustly identify the optimal number of clusters whose corresponding reconstructed spectra are the least similar to each other [34], but such additional methods are beyond the scope of this study. k-means is another traditional clustering algorithm that has been applied in studies in the realm of urban science [59–61]. We perform k-means clustering in the space of the first three PCs and compare to the results of hierarchical clustering, with results shown in Figure S5. We find that both approaches produce similar (if not exactly the same) results; since our goal is to compare a traditional clustering algorithm with SOMs, we choose to pursue our analysis using hierarchical clustering.

*2.5. Self-Organizing Maps*

The SOM method is similar to PCA in that both methods are used to reduce the dimensionality of a multidimensional dataset (i.e., income distributions for all census DAs) into a smaller set of characteristic modes (i.e., characteristic income distributions). While PCA decomposes data into a set of linearly independent modes that capture the most variance, the SOM method performs a non-linear projection from the input data space to a set of units (neural network nodes) on a 2-D grid or 'map.' For detailed descriptions and applications of the SOM algorithm, we refer the reader to [32,34,56,62]. Here, we discuss the main features of SOM as compared to PCA.

Characteristic income distribution patterns identified by the SOM method are analogous to the PCA modes of income distribution discussed in the prior section. While spatial information was contained in the PCs in PCA (i.e., PCs describe how strongly a given mode occurs in the income distribution of a given DA), the spatial information is encoded in the position on the 2-D SOM. SOMs are created through an iterative 'training' process, where each DA is projected onto a non-unique position in the SOM and each position in the SOM has an associated characteristic income distribution. The node to which a DA is mapped is called the 'best matching unit' (BMU). The spectrum of the BMU, as well as the spectra of its neighbouring nodes, are then adjusted to resemble the input sample more closely. The learning nature of the algorithm allows the network to evolve during the training stage. A key feature of the SOM method that is not captured through PCA is that a pattern for a given node on the 2-D map will resemble more closely the patterns from the neighbouring nodes on the map than nodes that are further apart. In this way, patterns that are situated at opposite corners tend to be the most different from each other.

Like for PCA, we are interested in how the spatial occurrence of characteristic income distributions change through time. We first create a SOM using the 2016 income distributions, which produces: (i) characteristic income distributions that are assigned to nodes on a 2-D map and (ii) a map of the city where each DA from 2016 is assigned a BMU. Next, we assign each DA's income distribution for 2022 to a node on the 2016 SOM by calculating the root-mean-squared-error (RMSE) between each 2022 income distribution and each characteristic income distribution in the SOM. The node whose pattern has the lowest RMSE is the BMU for that DA for 2022.

The SOM algorithm used in this study is adapted from the code developed by [62]. In brief, we use the open-source MATLAB-based SOM toolbox [63,64] to create a large SOM. We then perform PCA on the patterns from this large SOM and create a 'topography' defined as the sum of the squared first two PCs for each node. We choose the number of patterns as being the number of local maxima and global minimum in this topography. The local maxima represent the patterns that contain the most variance in the dataset. The global minimum represents the pattern that is least described by the first two modes and is therefore most different from the local maxima. Finally, we create a smaller SOM based on this number of nodes, in the shape with the most similar aspect ratio to the large SOM. We tested the sensitivity of our results by prescribing different sizes of SOMs but only present results for an 8-cluster (4 rows × 2 columns) map (see Section 3.2).

*2.6. Accessibility Analysis*

Catchment area analysis formed the basis for estimating the demographic make-up of people who could reach healthcare facilities from their homes within given travel-time thresholds. A detailed description of our method is provided in Reference [26] and we summarise this in Section 1.4 of the Supplementary Information. In summary, we used the open-source routing engine OpenTripPlanner [65] to calculate travel-time estimates between every pair of 'origin' and 'destination' (O-D) grid cells, by optimally combining walking and public transportation. We applied a modified version of the isochronic or cumulative-opportunity measure [25,66] to estimate the number of residents who could theoretically access healthcare facilities within a time threshold of 30 min. Acceptable travel-times vary depending on the travel mode, as well as socio-demographic and lifestyle factors, so we chose 30 min based on its use by the majority of metropolitan transport plans to assess accessibility via public transit [67]. Limitations of this approach are explored in more detail in Section 1.4 of the Supplementary Information.

In this longitudinal study, our analysis methods remained the same for 2016 and 2022. In other words, we assumed an unchanged distribution of facilities and transportation timetabling across our study period; only the population and income data were changed according to projections. We acknowledge that service provision is likely to change with future investments but without reliable data on planned infrastructure and timetabling changes, we assumed an unchanged transport network and healthcare provision in 2022. Instead, by isolating the effects of demographic and socioeconomic shifts on the size and composition of healthcare catchments, our analysis can serve as a baseline study of service demand to inform future work on changes to service provision and transportation capacity.

## 3. Results and Discussion

*3.1. Insights from Principal Component Analysis*

In this study, we use PCA and hierarchical clustering to determine the optimal number of cluster patterns to be prescribed in the SOM. As explained in Section 2.4 we use the demeaned, normalised continuous probability distributions for income in each grid cell as inputs for the PCA. The fraction of the total variance explained by each mode (Figure 1a,b) shows that the first three modes together explain >97% of variance in both the 2016 and 2022 income data. It is unsurprising that the first mode of our PCA captures such a high variance (~80%), given that income distributions tend to have relatively ubiquitous lognormal or Weibull shapes [51]. The remaining variance explained by modes other than the first three is likely attributable to contributions from noise.

The KDE of income in each cell is a linear combination of the modes shown in Figure 1e–g. The first mode is a close-to-normal distribution skewed towards lower incomes (Figure 1e), the second mode (depending on its sign) weights the distribution towards lower or higher incomes (Figure 1f) and the third mode (depending on its sign) either broadens or narrows the KDE (Figure 1g). The spatial distribution of the PCs associated with each of the three principal modes for the 2016 dataset are shown in Figure S1 in the Supplementary Information. These maps notably highlight the presence of negative PCs for mode 2 in the northwest and southeast of the city, which implies a higher prevalence of low-income household compared to the south and northeast.

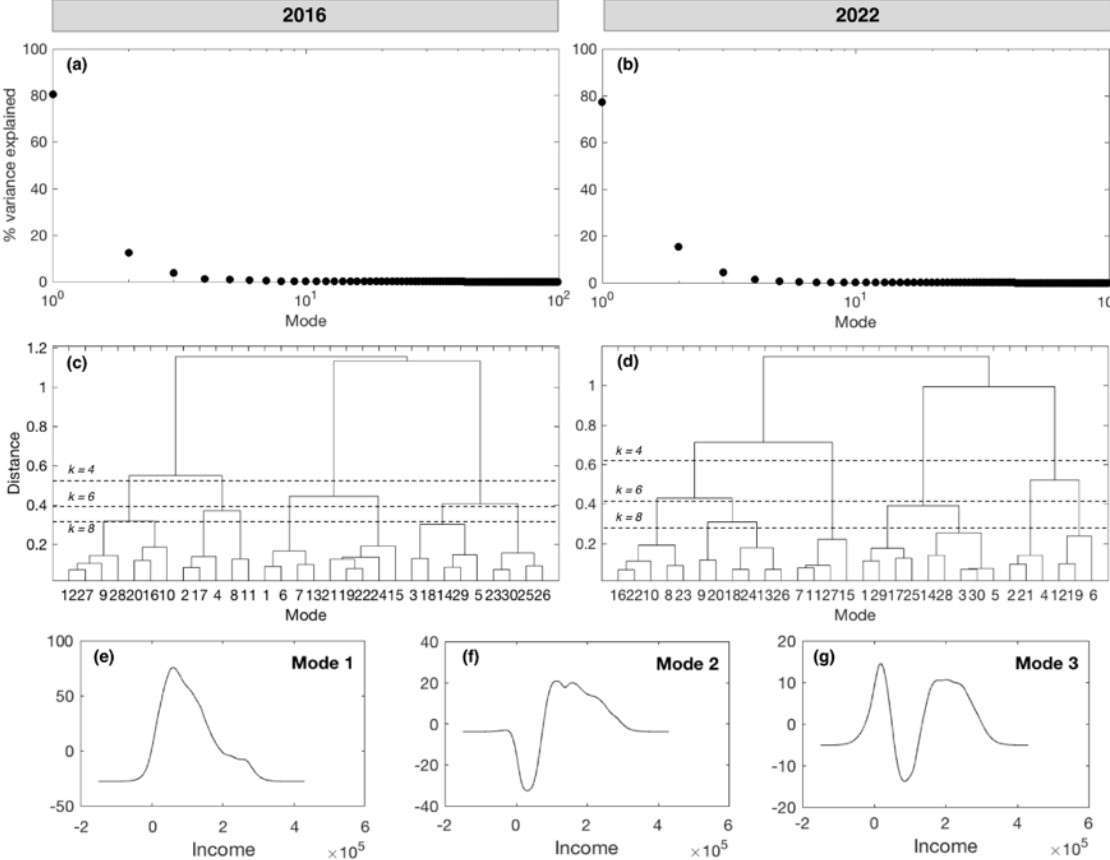

**Figure 1.** (**a**,**b**) Percentage of variance explained by the PC modes, for (**a**) 2016 and (**b**) 2022; (**c**,**d**) Dendrogram showing cluster structure and distances between clusters, for (**c**) 2016 and (**d**) 2022. Only ~35 subclusters are shown for ease of display. Vertical positions of the solid horizontal bars linking different clusters indicate the distance between those clusters. Dashed lines show possible cut off levels related to decreasing distances between clusters below; (**e**–**g**) First three PCA modes for 2016 income data (*y*-axis is unitless), which reveal the features that contain the most variance across each dataset.

The dendrograms presented in Figure 1c, d suggest that, for both the 2016 and 2022 datasets, a structure with $k = 4$–8 clusters represents a good compromise between minimising the number of clusters and maintaining relatively low intra-cluster variance. The spatially mapped PCA cluster topology for $k = 8$ clusters is shown in Figure 2a, b and the representative cluster patterns in Figure 2c. By 2022, clusters 5 and 6 spread across the city and cluster 1 declines. These shifts are reflected in the population proportions belonging to each cluster (Figure 2d). The median ages of each cluster show some change over the study period, particularly for clusters 5–8, where the median age increases by at least 4 years (Figure 1e). Clusters 5–8 tend to occur mostly in rural neighbourhoods, which suggests that elderly populations may be increasing in areas with generally poorer urban infrastructure.

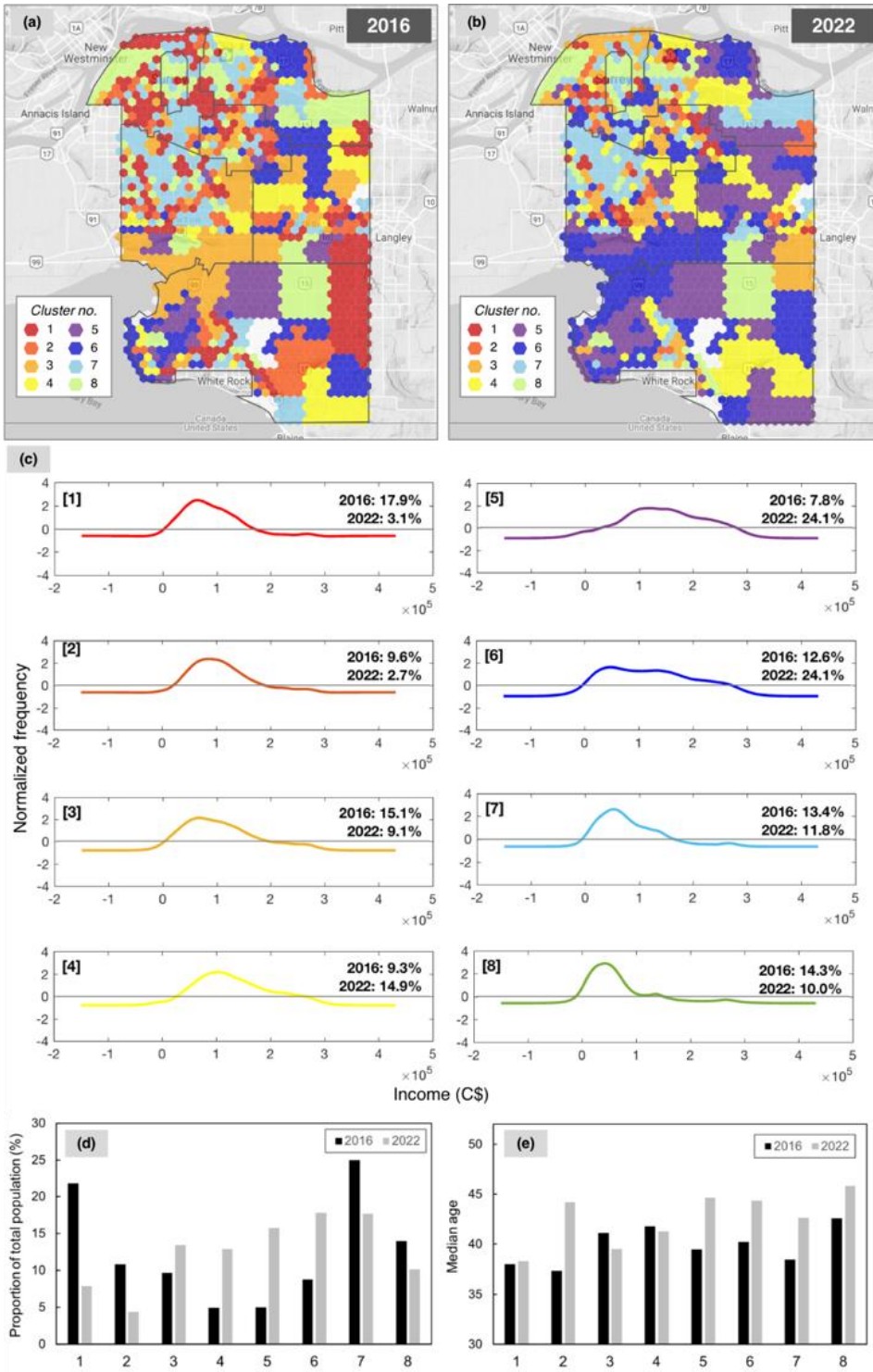

**Figure 2.** Spatially mapped PCA topology, coloured according to clustering, for (**a**) 2016 and (**b**) 2022. The modes and PCs were calculated using the 2016 dataset and the 2022 data were projected onto the 2016 modes to get the 2022 PCs. White cells show no data; (**c**) Most representative frequency distributions for each PCA cluster, showing in bold the frequency that they occur in the 2016 and 2022 maps. The *y*-axis on these plots varies about zero because the inputs are demeaned and normalised using their standard deviations; (**d**) Proportion of total city population belonging to each cluster, for 2016 And 2022; (**e**) Median age of population belonging to each cluster, for 2016 and 2022.

*3.2. Self-Organising Maps*

We now use SOMs to cluster and visualise our socio-economic data to relate them to accessibility metrics. In the following discussion, the term 'accessible' is used to refer to a facility being reachable by a resident from their home within 30 min using public transport/walking.

3.2.1. Relating SOM Topology to Accessibility and Travel-Time

Based on results from the PCA, which showed that 8 clusters ($k = 8$) provide good pattern diversity whilst retaining relatively low intra-cluster variance, we trained a SOM with a user-defined size of 4 rows $\times$ 2 columns. The spatially mapped SOM topologies for the 2016 and 2022 income data, as well as the characteristic frequency distributions, are shown in Figure S2 in the Supplementary Information. While it is useful to examine the distributions of the eight clusters separately, the advantage of SOM is that it provides the structure of the clusters relative to each other in the 2-D map. Since the SOM is stretched vertically, a duality exists between the top and bottom patterns of the map: cluster patterns 1 and 5 are most similar, as are cluster patterns 4 and 8. Rather than considering the eight clusters independently of each other in detail, we opt to group some of the clusters together to simplify our analysis. Since clusters 1 and 5 are characterised by close-to-normal income distributions, we group these into a 'high income' category (Figure S2). Conversely, clusters 4 and 8 have more positively skewed distributions, representing larger contributions from lower incomes, so we group these into a 'low income' category. In the grouped analysis, we do not consider clusters 2, 3, 6 and 7. This approach helps to highlight the most significant differences in terms of income patterns.

The spatially mapped SOM topology for the grouped clusters is shown in Figure 3a,b. In 2016, 27% of the city's grid cells are high income and 18% are low income, with a concentration of low income in the City Centre in the northwest. A similar proportion of the city's total population reside in high income (18%) versus low income (16%) areas (Figure 3c). By 2022, the coverage of high income increases significantly, sometimes incurring into previously low income areas. This finding is robust to whether the SOM is trained on 2016 data or on 2022 data (see Figure S3 in the Supplementary Information). As would be expected given the trend for an ageing population in Surrey, the median age for all clusters will increase by 2022 (Figure S2e), although low income areas will experience greater average ageing than high income areas (Figure 3d).

The SOM clusters provide a way of grouping the accessibility metrics calculated for each grid cell, which allows us to determine each cluster's aggregate accessibility to healthcare facilities (Figure 4). Whilst there is some significant variability in accessibility levels, most clusters will experience a rise in the average number of hospitals (Figure 4a) and walk-in clinics (Figure 4b) that can be reached by their residents across the study period. This occurs because many of the more rural, poorly connected members of each cluster transition to clusters 1 or 8 by 2022, thereby increasing the average accessibility of remaining members (while reducing the accessibility of cluster 8, for instance). When grouped, low income clusters have on average much better accessibility to hospitals (Figure 4c) and walk-in clinics (Figure 4d) than high income clusters. Increased accessibility between 2016 and 2022 is similarly observed for both low and high income groups (in the case of the low income group, increased accessibility in cluster 4 compensates for decreased accessibility in cluster 8). Specific inter-cluster transitions are examined in more detail in Section 3.2.4.

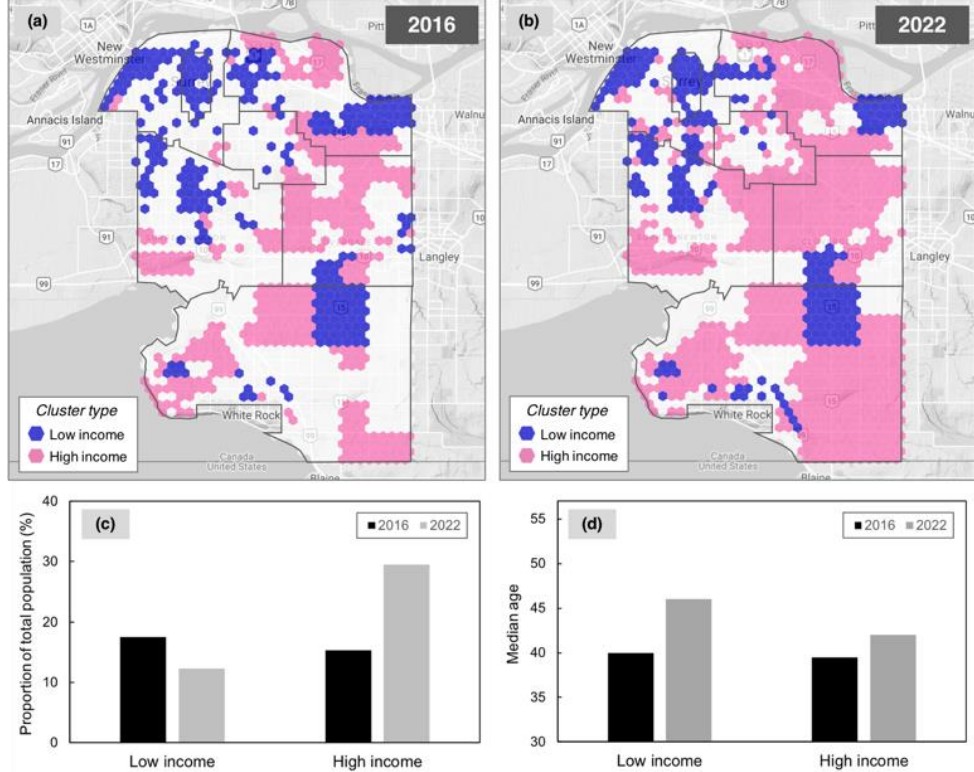

**Figure 3.** Spatially mapped SOM topology grouped according to income level (clusters 1 and 5 are considered high income, clusters 4 and 8 care considered low income, clusters 2, 3, 6 and 7 are excluded from the groupings), for (**a**) 2016 and (**b**) 2022. The 2016 data were used to train the SOM algorithm and this was used to classify both the 2016 and 2022 data. White cells show no data; (**c**) Proportion of total city population belonging to each income type, for 2016 and 2022; (**d**) Median age of population belonging to each income type, for 2016 and 2022.

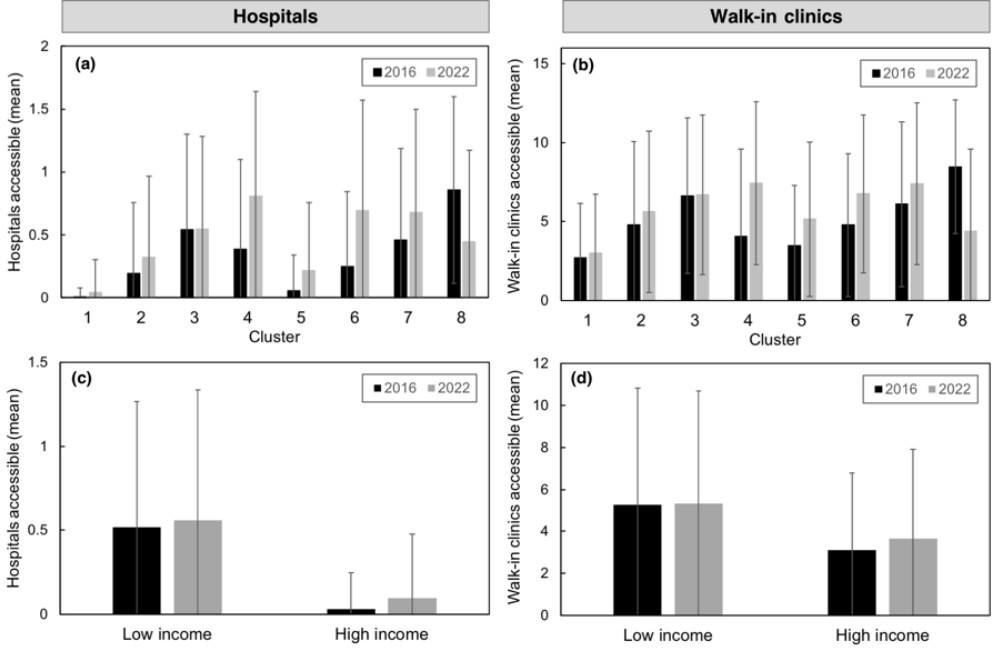

**Figure 4.** Mean number of (**a**) hospitals and (**b**) walk-in clinics accessible from each cluster's grid cells, and mean number of (**c**) hospitals and (**d**) walk-in clinics accessible from low- and high-income cluster groupings. Error bars show standard deviation.

Together, SOM cluster transitions and demographic shifts will have varied implications for accessibility for seniors. By 2022, ~12,000 more seniors will reside in areas with no access to a hospital (Figure 5a) and ~1500 more in areas with no access to a walk-in clinic (Figure 5b)—in both cases, most will be from a high-income cluster 1 area. There will be a large increase in the number of seniors living in high income clusters that have access to zero hospitals (Figure 5c). At the same time, over 15,000 more seniors will have access to at least 1 walk-in clinic, most residing in high income neighbourhoods (Figure 5d). This may put pressure on existing clinics for specific medical services, as demand will be concentrated in the few clinics available to neighbourhoods with high senior populations.

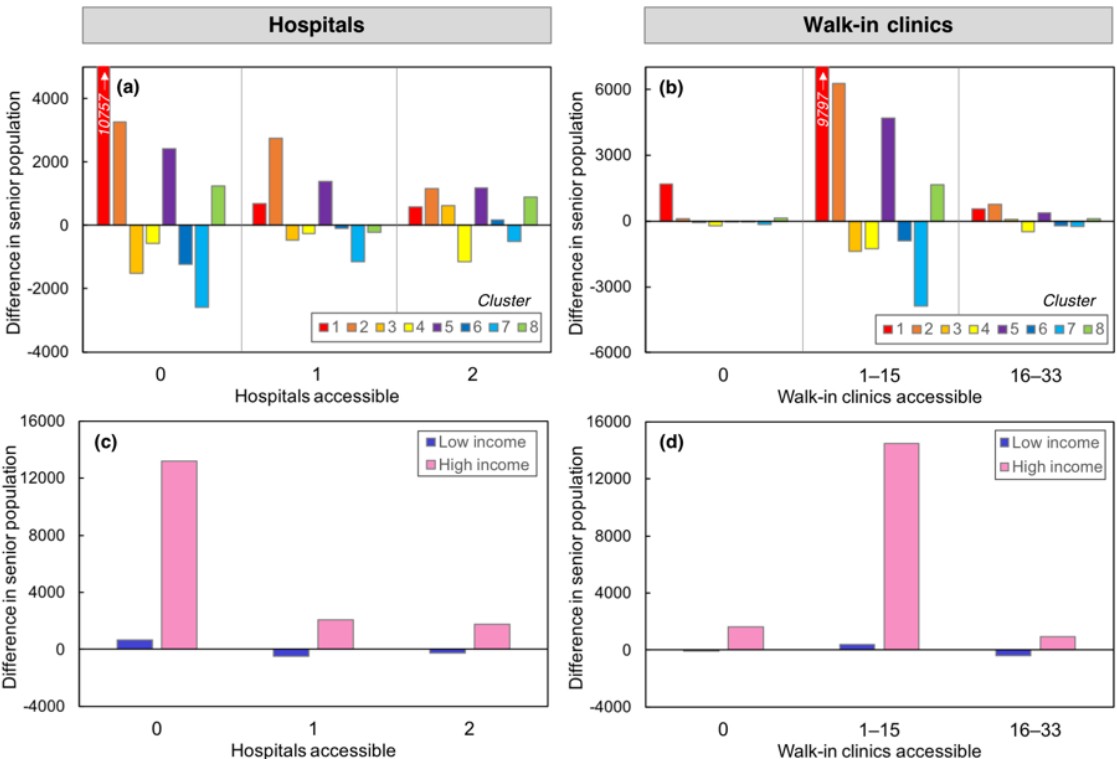

**Figure 5.** Difference (from 2016–2022) in the number of seniors with access to zero, up to half and up to all facilities, for (**a**) hospitals (all clusters), (**b**) walk-in clinics (all clusters), (**c**) hospitals (low vs. high income clusters) and (**d**) walk-in clinics (low vs. high income clusters). Vertical axes are curtailed to improve readability of graphs.

### 3.2.2. Attribution of Differences over Time

We now examine whether changes in accessibility over time are dominated by changes in the frequency of occurrence of each cluster pattern or by population change within each cluster pattern. For this we follow the approach of [62,68], who used cross-term analysis to detect the origin of differences between present and future climate phenomena. In our case, each of the eight cluster patterns for both 2016 and 2022 is associated with a certain population being able to access a given number of facilities. For a baseline year (i.e., 2016), the total population with access to a given number of facilities ($T$) across all clusters in the city ($P_{T,2016}$) can be expressed as:

$$P_{T,2016} = \sum_{n=1}^{N} f_n p_n \tag{3}$$

where $N$ is the total number of patterns in the SOM (i.e., $N = 8$); $f_n$ is the proportion of the population residing in each cluster; and $p_n$ is the average population with access to a given number of facilities,

per grid cell belonging to that cluster. To assess the magnitudes of the sources of differences between 2016 and 2022, $P_T$ in 2022 ($P_{T,2022}$) can be expressed as $P_{T,2016}$ plus the change in $P_T$, such that:

$$P_{T,2022} = \sum_{n=1}^{N} (f_n + \Delta f_n)(p_n + \Delta p_n) \tag{4}$$

where $\Delta f_n = (f_{n,2022} - f_{n,2016})$ and $\Delta p_n = (p_{n,2022} - p_{n,2016})$. Equation (4) can be expanded to give:

$$P_{T,2022} = \sum_{n=1}^{N} f_n p_n + f_n \Delta p_n + \Delta f_n p_n + \Delta f_n \Delta p_n \tag{5}$$

In Equation (4), the first difference term ($f_n \Delta p_n$) is the intra-pattern variability component, which reflects people of certain income levels (i.e., cluster membership) moving into or leaving the city. As an example, a positive intra-pattern component ($f_n \Delta p_n > 0$) could reflect residents moving into the city in 2022 with similar income distributions (i.e., cluster membership) as in 2016. The second difference term ($\Delta f_n p_n$) is the pattern frequency component, which reflects the change in distribution of people with certain income across the city, either as a result of movement within the city or of individuals getting richer/poorer over time. For instance, a negative pattern frequency component ($\Delta f_n p_n < 0$) indicates that, for the same population on average per cluster, more clusters experienced a decrease in total population proportion than experienced an increase. The third difference term ($\Delta f_n \Delta p_n$) is a combined term.

We perform this attribution of differences analysis for hospitals and walk-in clinics. In both cases, the analysis is run for populations with access to zero facilities ($P_0$) and for populations with access to one or more facilities ($P_{\geq 1}$), and separately for the city's senior population and the total (all ages combined) population (Table 1). For both populations, the intra-pattern variability component ($f_n \Delta p_n$) is the largest difference term when one or more healthcare facilities are accessible to residents ($P_{\geq 1}$). The intra-variability component is mostly negative, implying that residents will on the whole be moving into the city in 2022 with different income distributions compared with 2016 (e.g., higher-income individuals will move into previously lower-income areas). Conversely, when zero facilities are accessible ($P_0$), the pattern frequency component ($\Delta f_n p_n$) is the largest difference term for both the senior and total population. The positive values of this term could reflect people redistributing within the city to be in different neighbourhoods or experiencing changes in income (e.g., income drops after retirement in the case of seniors).

**Table 1.** Attribution of difference between 2016 and 2022 accessibility patterns. Net difference is difference in population between 2016 and 2022; this difference is decomposed into the intra-pattern variability component, pattern frequency component and combined term. The largest component in each case is highlighted in green.

| | Scenario | | Net Difference | Intra-Pattern Variability Component ($f_n \Delta p_n$) | Pattern Frequency Component ($\Delta f_n p_n$) | Combined Term |
|---|---|---|---|---|---|---|
| Hospitals | Access = 0 | Seniors | 11,775 | 5317 | 6031 | 426 |
| | | Total pop. | 27,575 | −16,454 | 47,389 | −3359 |
| | Access > 0 | Seniors | 5595 | 11,710 | −5650 | −464 |
| | | Total pop. | 16,495 | 56,801 | −44,311 | 4005 |
| Walk-in clinics | Access = 0 | Seniors | 1429 | −110 | 1392 | 148 |
| | | Total pop. | 2620 | −6091 | 10,277 | −1565 |
| | Access > 0 | Seniors | 15,941 | 17,137 | −1011 | −186 |
| | | Total pop. | 41,450 | 46,438 | −7199 | 2211 |

These results suggest that in areas with no current access to hospitals/walk-in clinics, accessibility changes resulting from the (re)distribution of residents–or from residents getting poorer or richer–will dominate over changes resulting from people moving into or out of the city. The opposite is true

for areas with access to at least one healthcare facility. From a policy and transport planning perspective, this implies that municipalities may have to consider healthcare accessibility equity through differentiated lenses: in neighbourhoods with existing access to healthcare, population growth will likely cause service bottlenecks and an increase in facility provision may be necessary. In poorly connected neighbourhoods, on the other hand, tackling the effects of income change (both positive and negative) may require adapting public transportation infrastructure and timetabling. This will be particularly important in areas where projected senior population growth is highest and therefore private vehicle usage will likely decrease.

### 3.2.3. Accessibility to Nearest Facility

Quantifying how quickly residents can reach the nearest facility is a complementary measure of accessibility. The cumulative frequency distributions of travel-time to the closest hospital and walk-in clinic for the grouped clusters are presented in Figure 6 (individual distributions for all clusters are shown in Figure S4 in the Supplementary Information). High income clusters have significantly longer travel-times than low income clusters to hospitals (Figure 6a) and walk-in clinics (Figure 6b). From 2016 to 2022, both low and high income clusters experience a decrease in travel-time to the nearest facility (i.e., a leftward shift in the graphs). The shift towards lower cumulative travel times implies that facilities will on average become more accessible to residents, increasing pressure on services at specific healthcare locations. At the same, time, additional transit routes or more frequent timetabling may be required to service people moving into more rural areas of the city (e.g., Figure S4f shows how cluster 8 (low income) will suffer from far longer travel times to clinics by 2022).

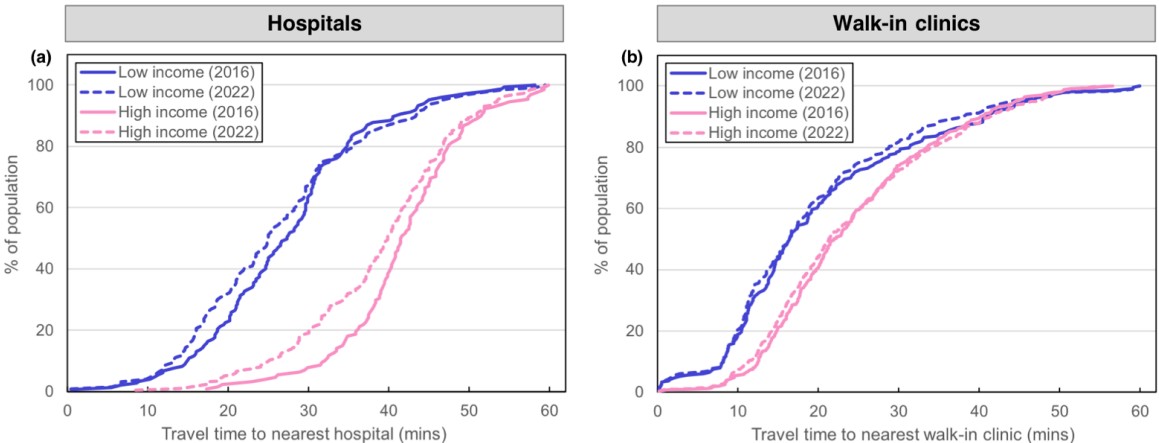

**Figure 6.** Cumulative frequency distributions of travel-time to closest (**a**) hospital and (**b**) walk-in clinic for each cell in 2016 and 2022, grouped into low (clusters 4 and 8) and high (clusters 1 and 5) income.

### 3.2.4. Cluster Change

Mapping the pathways of change between different clusters reveals how BMU membership is projected to shift through time (Figure 7). Cluster 1 dominates by 2022 for two main reasons: firstly, 99% of cells that were originally in cluster 1, remain in cluster 1, and secondly, cluster 1 absorbs a large proportion of cells from clusters 2 and 5. Membership to clusters 6 and 7 drops significantly by 2022 due to their cells transitioning to clusters 2 and 5. No transitions exceeding one neighbouring patterns was observed (except for 7 to 5), suggesting that neighbourhoods are not projected to experience rapid changes in income distribution. Notably, none of the cluster rows move 'down the map'—in other words, no significant transitions from high to low income are observed across the city. The interpretation of our cluster map is facilitated by the topological nature of the SOM: since clusters that are close on the SOM are more similar than clusters that are far apart, the movement of clusters 'up the map' indicate a characteristic change from lower to higher income across Surrey.

This visualization highlights the benefits of using SOMs as tools for communicating change over time to non-specialists such as policymakers.

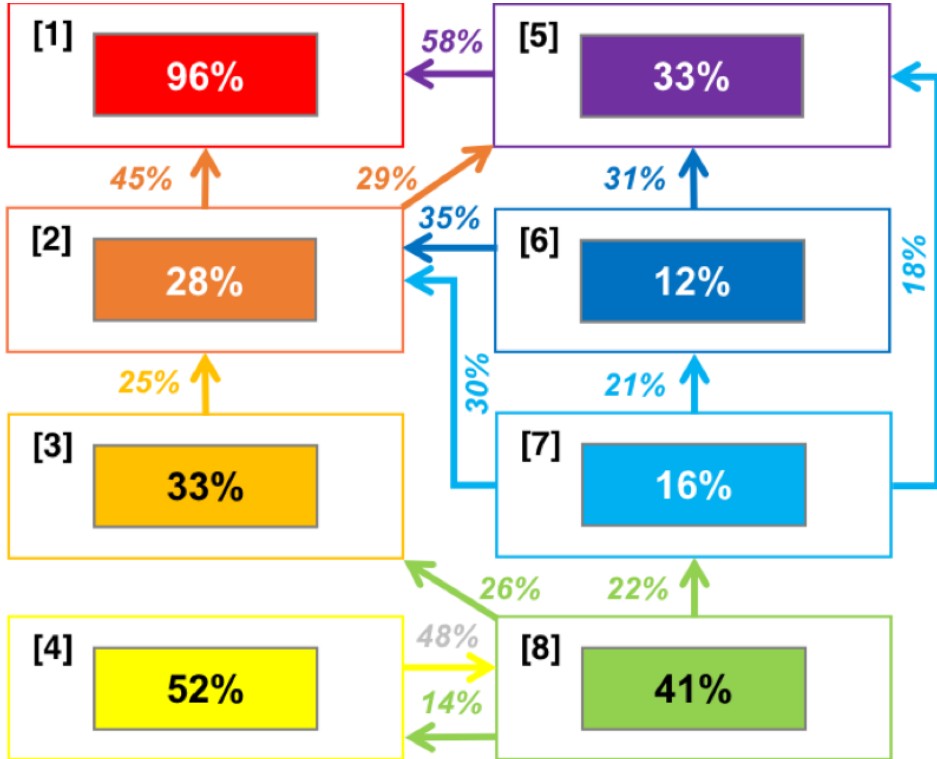

**Figure 7.** Pathways of cluster changes between 2016 and 2022. Value at the centre of each box corresponds to the proportion of the cluster's population that remained the same between 2016 and 2022. Coloured arrows show the proportion of the population of a given cluster in 2016 that transitioned to a new cluster in 2022 (e.g., 45% of all residents belonging to cluster 2 in 2016 changed to cluster 1 in 2022). Pathways with frequencies <10% are not shown.

We conducted 'cluster distance' analysis in order to quantify the degree to which the BMU membership of individual grid cells changed by between 2016 and 2022. To calculate cluster distance, we first performed PCA on the distributions of the eight BMUs shown in Figure S2 and selected the first two modes (which explain 93% of total variance) and their associated PCs for analysis. If a grid cell changed BMU between 2016 and 2022, we quantified the magnitude of that change as the Euclidean distance in PC space between the 2016 cluster pattern ($PC1_{2016}$, $PC2_{2016}$) and the 2022 cluster pattern ($PC1_{2022}$, $PC2_{2022}$). For grid cells whose BMU did not change during the study period, the cluster distance was set to zero.

The median cluster distance for cells with different levels of accessibility to hospitals and walk-in clinics is shown in the form of box-and-whisker plots in Figure 8a,b. Whilst the interquartile ranges are relatively large in all cases, the median cluster distance is lowest for cells with the poorest (i.e., 0) and the best (i.e., 2) access to hospitals (Figure 8a). In contrast, median cluster distance is lowest for cells with an intermediate level of accessibility to walk-in clinics (i.e., 1–15) (Figure 8b). This supports the notion that the most significant changes in income distribution will occur in areas with currently very good or very poor access to clinics.

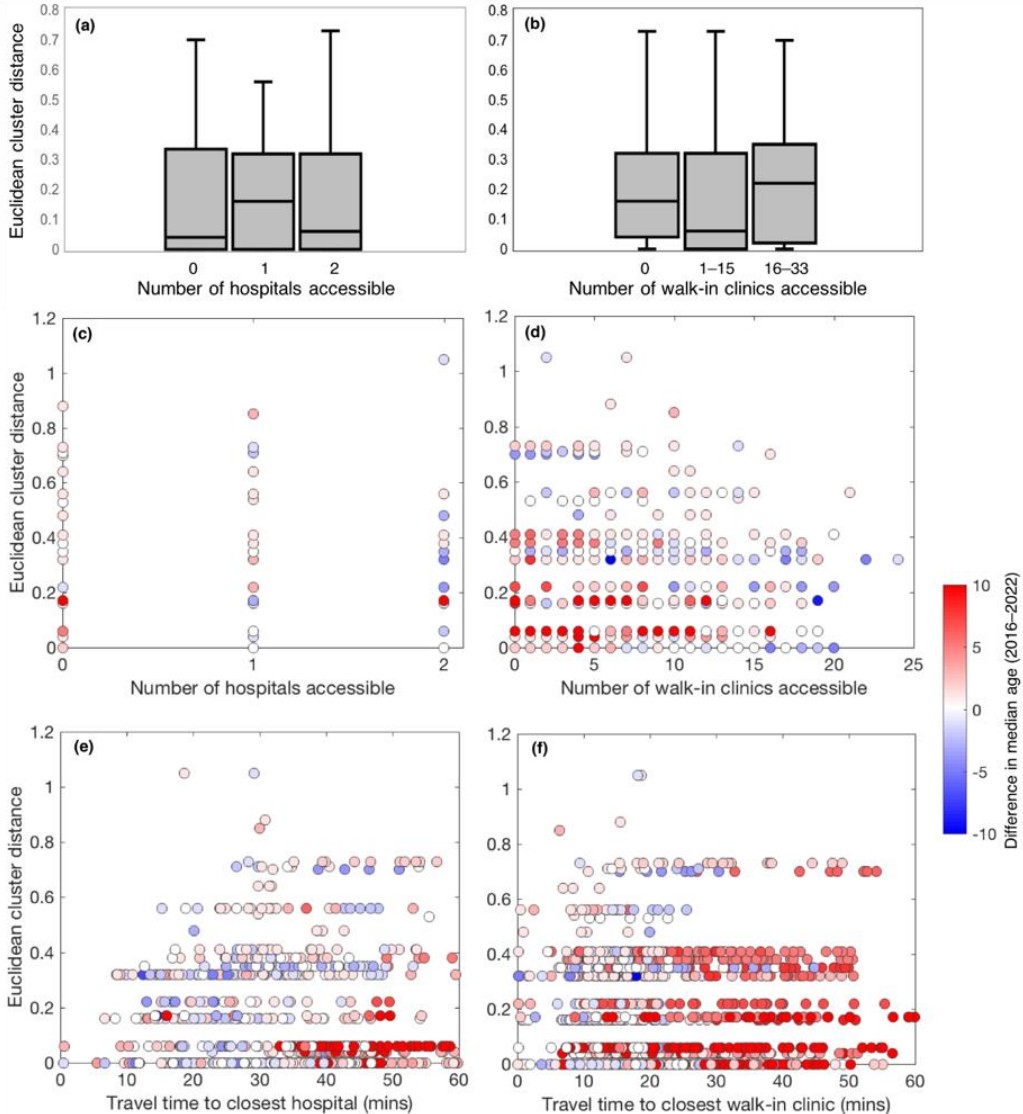

**Figure 8.** (**a**,**b**) Box-and-whisker plots of cluster distance, grouped by level of accessibility, to (**a**) hospitals and (**b**) walk-in clinics; (**c–f**) Scatterplots showing relationships between access to healthcare facilities and the Euclidean cluster distance for each cell, with a third (coloured) variable showing the difference in median age between 2016 and 2022.

To explore this in further demographic detail, we present in Figure 8c–f the relationships between accessibility to healthcare facilities, Euclidean cluster distance and difference in median age between 2016 and 2022. Difference in median age is used as an indicative measure of ageing; an increase in median age could be the result of an influx of older residents but also from a proportional decrease in young people, as has been observed across Surrey over the past two decades [46]. For hospitals (Figure 8c) and walk-in clinics (Figure 8d), the largest increases in median age (i.e., redder colours) are projected to occur in areas with few accessible facilities and lower cluster distances. This implies that ageing populations will be concentrated in areas with relatively poor hospital and clinic accessibility, although income distributions in these areas will remain relatively stable. In contrast, median age will decrease (bluer colours) in areas experiencing large changes in income distribution and/or with access to numerous healthcare facilities. A similar picture emerges from the relationships with travel time to the closest facility (Figure 8e,f), with populations that age most being located furthest away from the closest hospital and clinic, where changes in income distribution are small. Populations that are

getting younger appear to dominate relatively stable clusters with shorter travel times or unstable clusters with longer travel times (particularly to hospitals).

These findings could have critical ramifications for healthcare accessibility equity for urban centres like Surrey. By 2022, over 15,000 more seniors are projected to move into areas with access to at least 1 walk-in clinic and ~3,000 more seniors into areas with access to both of Surrey's hospitals. The highest rates of population ageing are projected to occur in neighbourhoods where healthcare access is poor, many of which are lower-income. This could create a dual accessibility problem with regards to the senior population: first, pressure may increase on existing facilities for certain specialised services as large numbers of seniors move into their catchments, and second, lower-income seniors will increasingly reside in areas poorly connected to healthcare services, particularly rural suburbs.

## 4. Conclusions

Powerful machine learning (ML) techniques are important tools for robustly analysing increasingly large urban datasets. As we have shown in this study, ML methods such as self-organising maps (SOMs) provide robust ways to uncover underlying data patterns and to use large multivariable datasets for recommending actionable policy interventions. We used SOMs to characterise income distributions in the City of Surrey, to map how they will change in space and time and to relate them to accessibility metrics for healthcare services. The performance of the SOM was compared to the output from a combination of principal component analysis (PCA) and hierarchical clustering, to determine the optimal number of clusters needed in the final map.

Our results suggest that rapidly changing urban centres such as Surrey could soon suffer from a dual accessibility problem. First, large senior populations will move into areas with very good access to clinics, putting pressure on existing facilities. Second, lower-income seniors will increasingly reside in areas poorly connected to healthcare services. These dynamics raise important questions about transport mode choice, since higher-income seniors may be able to somewhat mitigate the impacts of low accessibility to healthcare through owning and using private vehicles, whilst lower-income senior residents are likely to be highly reliant on public transportation. We did not include private vehicles in our analysis, so future research should integrate mode choice to improve our understanding of accessibility equity issues in multi-modal transportation networks.

While our study is focused on the City of Surrey, fertility rates are declining and life expectancy is rising in many cities around the world [69]. Resulting shifts to more elderly, and thus less mobile, populations will pose major challenges for ensuring healthcare access equity [70–72]. Urban (re)development cannot always take place in areas well served by public transportation, so accessibility to essential services could suffer amongst specific marginalised groups, particularly senior or low-income populations. From a policy and transport planning perspective, it may be necessary to focus not only on magnitudes of population change, but also on improving infrastructure and timetabling of public transportation to account for shifts in the socio-demographic makeup of certain neighbourhoods.

This study focused on analysing changes in demographics and accessibility demand, so the scope and lack of appropriate data meant we had to assume an unchanged transport network and provision of facilities. However, the approach we introduce here could be used to inform different infrastructure investment strategies, including transportation and health facilities planning. Methods such as agent-based modelling (ABM), which account for diverse socio-behavioural decision-making rather than assuming fixed behaviours [73] would complement our combined clustering method for more nuanced accessibility analyses for specific case studies. There is great potential in the judicious deployment of ML methods within urban analytics.

**Supplementary Materials:** The following are available online at http://www.mdpi.com/2413-8851/3/1/33/s1, Figure S1: Spatial distribution of principal components; Figure S2: Spatially mapped SOM topology, coloured according to clustering; Figure S3: Spatially mapped SOM topology; Figure S4: Cumulative frequency distributions of travel-time to closest facility for each cell; Figure S5: Comparison of cluster results when using k-means clustering and hierarchical clustering.

**Author Contributions:** J.R.M. and S.A. were responsible for conducting the data analysis. S.A. and V.R. developed the code for the self-organizing maps and principal component analysis and J.R.M. developed the code for the catchment area analysis. All four authors contributed to the design of the study and writing the manuscript.

**Funding:** This research was funded by the Pacific Institute for Climate Solutions (PICS), Cascadia Urban Analytics Cooperative (CUAC), NSERC Canada Graduate Scholarship and NSERC Discovery Grant.

**Acknowledgments:** We thank the Pacific Institute for Climate Solutions (PICS) and Cascadia Urban Analytics Cooperative (CUAC) for partly funding this research for J.R.M. and M.T. S.A. was funded by an NSERC Canada Graduate Scholarship and V.R. was funded by an NSERC Discovery Grant. We thank three anonymous reviewers whose comments helped to improve this paper.

**Conflicts of Interest:** The authors declare no conflict of interest in this work.

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
