# Peer review of "Insights from Self-Organizing Maps for Predicting Accessibility Demand for Healthcare Infrastructure"

_urbansci, doi:10.3390/urbansci3010033_

Round 1

Reviewer 1 Report

The paper makes a good effort in trying to use principal component analysis (PCA) and self- organising map (SOM) over socio-economic factors high resolution income data for 2016 and 2022 to Predict Accessibility Demand for Healthcare Infrastructure. This is an interesting research project to me, but I still have the following concerns about this manuscript. First, the authors need to significantly improve the literature review on this subject to give the audience more thorough understanding on the progress of related studies so far. Second, the PCA and SOM employed in this research are interesting, but I don’t really understand why finally these two approaches were selected for this research, which need to be justified with details in this manuscript. Furthermore, the comparison of the employed approaches and other similar approaches also need to be conducted in both literature review and (preferably) the experiments. Third, the selection of the size of hexagonal cell need to be explained more clearly and discussed. The uncertainty of datasets employed, such as OpenStreetMap, also needs to be discussed. Lastly, the writing, tables and maps also need to be improved.

Author Response

The paper makes a good effort in trying to use principal component analysis (PCA) and self- organising map (SOM) over socio-economic factors high resolution income data for 2016 and 2022 to Predict Accessibility Demand for Healthcare Infrastructure. This is an interesting research project to me, but I still have the following concerns about this manuscript.

First, the authors need to significantly improve the literature review on this subject to give the audience more thorough understanding on the progress of related studies so far.

This is quite an unhelpful comment as there is no detail whatsoever as to what constitutes ‘significantly improve’. We have added some detail to the literature review linking accessibility to healthcare: ‘Poor healthcare accessibility is known to result in lower healthcare utilization and inferior health outcomes (Hiscock et al., 2008; Dai, 2010; Kawakami et al., 2011), so determining where to build new healthcare facilities is an ongoing challenge for urban planners. Moreover, declining fertility rates and increased longevity are resulting in older, less mobile populations with different accessibility needs compared to younger, more mobile ones (Neutens, 2015; Frank et al., 2019). A major issue for growing cities is to ensure that their transportation systems remain inclusive and equitable across different population groups (Wegener, 2013).’ (p.2)

We now give more detail about the applications of SOMs to other academic fields: ‘SOM algorithms have been applied to a wide range of disciplines (see reviews in Johnsson, 2012), from analysis of volcano seismic spectra (Unglert et al., 2016) and atmospheric aerosol tracking (Ashpole & Washington, 2013) to land value prediction (Sohn, 2013). Other ML techniques (e.g. neural networks, random forest classifiers) have also been applied to urban transportation problems, including for predicting traffic flows (Fusco et al., 2016) and travel mode choice (Hagenauer & Helbich, 2017), and for clustering transit card usage (Mahrsi et al., 2017).’ (p.3)

We also specify more detail about our SOM methodology variously throughout the Methods and Discussion sections (see our responses to related comments from the other Reviewers).

Second, the PCA and SOM employed in this research are interesting, but I don’t really understand why finally these two approaches were selected for this research, which need to be justified with details in this manuscript.

We feel that we already provide substantial detail as to why the PCA and SOM approaches were used in this particular research. As part of our explanation of SOM in the Introduction, we write that, ‘SOM analysis has had a relatively limited impact in the social sciences, despite the possibilities it offers to intuitively visualise complex socio-demographic patterns, which can notably be helpful in the context of policy-making.’ (p.3)

At the end of our Introduction, in our statement of aims, we note that: ‘We use both approaches to reduce the dimensionality of our income datasets and draw out patterns in the underlying data, to then relate them to accessibility metrics. While both PCA and SOM are established methodologies in themselves, our combination of classical and modern data analysis methods provides novel, detailed insights into a city’s evolution in terms of population growth, income distribution and accessibility to essential services.’ (p.4)

We also note in our Methods: ‘Principal component analysis (PCA) is a well-established classical data analysis method for reducing dimensionality of a high-dimensional dataset.’ (p.6)

The entirety of section 2.6 is devoted to explaining SOM and how it relates (and differs) to the more traditionally-employed PCA.

In section 3.2.4 (‘Cluster change’), we note that the results discussed in Figure 7 were simplified and visualized by the use of SOMs, highlighting the method’s strength as a data visualization tool: ‘The interpretation resulting from our cluster mapping is facilitated by the topological nature of the SOM: since clusters that are close on the SOM are more similar than clusters that are far apart, the movement of clusters ‘up the map’ indicate a characteristic change from lower to higher income across Surrey. This visualization highlights the benefits of using SOMs as tools for communicating change over time to non-specialists such as policymakers.’ (p.20)

In our Conclusions, we further note that, ‘ML methods such as self-organising maps (SOMs) provide robust ways to uncover underlying data patterns and to use large multivariable datasets for recommending actionable policy interventions […] The performance of the SOM was compared to the output from a combination of principal component analysis (PCA) and hierarchical clustering, to determine the optimal number of clusters needed in the final map.’ (p.23)

Furthermore, the comparison of the employed approaches and other similar approaches also need to be conducted in both literature review and (preferably) the experiments.

We now discuss: ‘k-means is another traditional clustering algorithm that has been applied in studies in the realm of urban science (Fazlollahi et al., 2014; Hooper et al.,2015; Liu et al., 2018). We perform k-means clustering in the space of the first three PCs and compare to the results of hierarchical clustering, with results shown in Supplementary Figure 5. We find that both approaches produce similar (though not exactly the same) results, but since our goal is to compare a traditional clustering algorithm with SOMs, we choose to continue our analysis with hierarchical clustering.’ (p.7–8)

Third, the selection of the size of hexagonal cell need to be explained more clearly and discussed.

We now specify, ‘The hexagonal shape of the cell was chosen to reduce sampling bias from edge effects, and the size of the grid cell was designed to provide enough granularity in our spatial data while roughly matching the average size of a DA in Surrey. This helped to minimise downscaling errors when assigning census data to hexagonal cells. We followed the method of Mayaud et al. (2018) to assign the census data to each hexagonal cell (see section 1.1 of the Supplementary Material for a brief summary).’ (p.4–5)

The uncertainty of datasets employed, such as OpenStreetMap, also needs to be discussed.

We accept that the uncertainty of the OSM dataset requires attention in our manuscript. We therefore add: ‘OSM is a prominent example of volunteered geographic information, where users contribute voluntarily to its open-source development. As such, some questions have arisen about its accuracy in many parts of the world. In a global analysis of the ‘completeness’ of OpenStreetMap, Barrington-Leigh & Millard-Ball (2017) found that Canada has a fully mapped street network, so we consider OSM to be reliably complete for our analyses.’ (p.5)

Lastly, the writing, tables and maps also need to be improved.

We consider this comment to provide no substantive detail upon which we can make productive improvements.

Reviewer 2 Report

The study presents a strategy to analyze the changes in demography and the demand for accessibility in urban areas. For this, the authors use techniques based on machine learning and organizing maps (SOMs) to discover underlying data patterns and then recommend policy interventions. In general, the document is well structured and written. The results validate the proposal done. I consider that the work can be accepted with minor changes that I describe below:

1. At the end of the introduction, I consider it necessary to add a paragraph indicating the structure of the document to be followed. For example (In section ii  describes ..., in section 3 ...)
2.- Could you expand the use of the KDE-based method? There are other options and why are not others?
3. In line 159 Why are you using only the first few? And ¿All dataset?

Author Response

The study presents a strategy to analyze the changes in demography and the demand for accessibility in urban areas. For this, the authors use techniques based on machine learning and organizing maps (SOMs) to discover underlying data patterns and then recommend policy interventions. In general, the document is well structured and written. The results validate the proposal done. I consider that the work can be accepted with minor changes that I describe below:

1. At the end of the introduction, I consider it necessary to add a paragraph indicating the structure of the document to be followed. For example (In section ii  describes ..., in section 3 ...)

We have added the following paragraph at the end of our Introduction: ‘The remainder of this paper is presented in three parts. The next section introduces the socio-economic context of our case study, outlines the data sources used and explains the use of PCA and SOM in this study. Section 3 presents our analysis and a discussion of the results. Section 4 summarizes the main conclusions of this paper.’ (p.4)

2.- Could you expand the use of the KDE-based method? There are other options and why are not others?

In Section 2.3., we have edited the following sentence to emphasize that KDEs are a well-established technique with widespread use, and we provide further references that explore their properties.  We believe this is sufficient justification for their use in our study. Edited sentence: ‘We use MATLAB’s built-in ‘ksdensity’ kernel smoothing method to perform a kernel density estimation (KDE), which is a well-established technique for estimating probability density functions (Rosenblatt, 1956; Parzen, 1962; Silverman, 1986; Scott, 1992).’

3. In line 159 Why are you using only the first few? And not all dataset?

We already specify in the text that we reconstruct the original dataset ‘using only the first few (most important) modes, ensuring that a majority of the variance is accounted for whilst filtering out noise.’ This is common practice in the SOM methodology – including all the modes would be counter-productive for clustering purposes.

Reviewer 3 Report

This paper studied on the insights by combining PCA and SOM for prediction of accessibility. The proposed method is well explained in the manuscript, in addition, the experimental results show the validity of the research.

Author Response

This paper studied on the insights by combining PCA and SOM for prediction of accessibility. The proposed method is well explained in the manuscript, in addition, the experimental results show the validity of the research. 

We thank the Reviewer for their positive comments.

Reviewer 4 Report

The article proposes an evolution of some metrics for the analysis of the accessibility of health care in smarter cities. This subject is interesting. In relation to the great challenges, the approach adopted needs to be improved in the literature review of smart cities and their challenges. The attempt to propose analyzes for the redesign of health care needs to be connected with the intelligent health driver.

General point:

Point 1. Has the article received a format revision by the MDPI team?

Point 2. The approach adopted needs to be improved in the literature review of smart cities and their challenges.

Author Response

The article proposes an evolution of some metrics for the analysis of the accessibility of health care in smarter cities. This subject is interesting. In relation to the great challenges, the approach adopted needs to be improved in the literature review of smart cities and their challenges. The attempt to propose analyzes for the redesign of health care needs to be connected with the intelligent health driver.

We thank the Reviewer for their comments and for pointing out the missing link between our aim to introduce new statistical techniques for analysis, and the broader drive towards making cities ‘smarter’. We address this point in more detail below.

General point:

Point 1. Has the article received a format revision by the MDPI team?

We have received a separate format revision from MDPI, which we have integrated in our revised manuscript.

Point 2. The approach adopted needs to be improved in the literature review of smart cities and their challenges.

To better link the challenges faced by growing ‘smart’ cities to accessibility challenges, we now add a paragraph to our Introduction: ‘Disparities in healthcare access arise because of a variety of physical and socioeconomic factors, including social status, household income, transport infrastructure and the spatial distribution of cities (Hiscock et al., 2008; Bissonnette et al., 2012), but it is difficult to conclusively isolate causation within complex urban fabrics (Hickford et al., 2015). Poor healthcare accessibility is known to result in lower healthcare utilization and inferior health outcomes (Hiscock et al., 2008; Dai, 2010; Kawakami et al., 2011), so determining where to build new healthcare facilities is an ongoing challenge for urban planners. Moreover, declining fertility rates and increased longevity are resulting in older, less mobile populations with different accessibility needs compared to younger, more mobile ones (Neutens, 2015; Frank et al., 2019). A major issue for growing cities is to ensure that their transportation systems remain inclusive and equitable across different population groups (Wegener, 2013).’ (p2–3)

Round 2

Reviewer 4 Report

The article had a substantial improvement in its content and presentation form but despite the improvements, the theoretical framework needs to be expanded, an approximate number of 80 updated citations (from the last five years and with DOI) is ideal. There are currently 58 citations. Especially about Smart Cities drivers.

Author Response

We have taken the reviewer's comments on board and added 14 more references in our Introduction, to expand in detail on the concepts behind smart cities and how they relate to accessibility and equity to services. We trust this satisfies the reviewer's request for more citations.

We now write in the Introduction: "Accessibility (the ease with which people can reach places or opportunities) is an important indicator of the liveability and sustainability of cities. In the global drive to make cities ‘smarter’, there has been an observable trend for greater urban interconnection, especially through investments in physical infrastructure and information communication technologies (ICT) [1,2]. Innovative networks of connected devices and sensors, coupled with smart applications and data analytics, have also revolutionised the ways in which a variety of actors – from city governments to citizens – can simultaneously optimise existing systems [3], improve the quality of life urban residents [4–5] and address sustainable development needs [6–8]. There are widespread calls for smart cities to ensure their services remain inclusive and equitable as they grow [9–10], because evidence shows that universal access to basic services positively impacts human development and societal progress [11–14]."